# Private Sketches for Linear Regression

## Abstract

Linear regression is frequently applied in a variety of domains, some of which might contain sensitive information. This necessitates that the application of these methods does not reveal private information. Differentially private (DP) linear regression methods, developed for this purpose, compute private estimates of the solution. These techniques typically involve computing a noisy version of the solution vector. Instead, we propose releasing private sketches of the datasets, which can then be used to compute an approximate solution to the regression problem. This is motivated by the *sketch-and-solve* paradigm, where the regression problem is solved on a smaller sketch of the dataset instead of on the original problem space. The solution obtained on the sketch can also be shown to have good approximation guarantees to the original problem. Various sketching methods have been developed for improving the computational efficiency of linear regression problems under this paradigm. We adopt this paradigm for the purpose of releasing private sketches of the data. We construct differentially private sketches for the problems of least squares regression, as well as least absolute deviations regression. We show that the privacy constraints lead to sketched versions of regularized regression. We compute the bounds on the regularization parameter required for guaranteeing privacy. The availability of these private sketches facilitates the application of commonly available solvers for regression, without the risk of privacy leakage.

## 1 Introduction

Differentially private (DP) linear regression methods aim to compute the solution to the problem privately. Given a data matrix $X \in \mathbb{R}^{n \times d}$ and a response vector $y \in \mathbb{R}^n$, the linear regression problem is to compute $\beta^* = \underset{\tilde{\beta} \in \mathbb{R}^d}{\operatorname{argmin}} ||X\tilde{\beta} - y||_p^q$, where $p = q = 1$ denotes $\ell_1$ (least absolute deviations) regression and $p = q = 2$ denotes $\ell_2$ (least squares) regression. DP linear regression algorithms estimate the solution $\beta^*$ privately. Most of the literature on DP linear regression focusses on $\ell_2$ regression, while Wang & Xu (2022); Liu et al. (2024) proposed algorithms for DP $\ell_1$ norm regression.

Since the solution for $\ell_2$ regression has a closed form expression, $\beta^* = (X^\top X)^{-1} X^\top y$, existing approaches include computing private estimates of $X^\top X$ and $X^\top y$ Wang (2018), using random projections for the privately estimating $\beta^*$ Sheffet (2017), estimating the solution using private gradient descent type algorithms Varshney et al. (2022), as well as exponential mechanism type approaches Liu et al. (2021). Brown et al. (2024) proposes estimating the covariance matrix on a "good" set of data (discarding points with high leverage scores as well as high residuals) and perturbing the least squares estimate. For the case of $\ell_1$ regression, Liu et al. (2024) proposes an iterative algorithm for estimating the solution which includes a kernel density estimation step for the purpose of introducing noise. Wang & Xu (2022) proposes using the exponential mechanism for computing the solution privately.

Each of these methods, except Sheffet (2017), involve multiple steps for estimating some property of the data matrix, which is then used for making the solutions private. In this paper, we propose releasing differentially private sketches of the data for solving the linear regression problem. We consider the classic *sketch-and-solve* paradigm (Woodruff, 2014) where an approximate solution to the regression problem is computed on a sketch of the data matrix. The sketched data matrix has much fewer rows than the original data matrix due to which computing the solution on the sketch is computationally efficient. Various sketching algorithms exist

that are able to given a good approximation for both $\ell_1$ and $\ell_2$ regression problems Sarlos (2006); Nelson & Nguyên (2013); Meng & Mahoney (2013); Woodruff (2014); Clarkson & Woodruff (2017); Munteanu et al. (2021; 2023). While sketching has been utilized for tasks such as private low rank approximation Upadhyay (2017) and distributed private data analysis Burkhardt et al. (2025), to the best of our knowledge, this is the first work that has looked at releasing private sketches for $\ell_1$ and $\ell_2$ regression. We note that Sheffet (2017) utilizes random projections of the data matrix for computing the solution to $\ell_2$ regression privately.

An advantage of releasing private sketches of the data matrix is that the sketched regression problem can be solved using readily available solvers for both $\ell_1$ and $\ell_2$ regression, instead of having to estimate various properties of the data. Additionally, these private sketches can be queried infinitely, without any loss of privacy. The idea of releasing private sketches is closely related to the idea of releasing private coresets, which has been done for clustering (Feldman et al., 2009; 2017). In this paper, we have not looked at private coresets for linear regression since coresets for regression generally depend on importance sampling from a data dependent distribution and are sensitive to changes in the dataset.

One of the most widely used sketching algorithms for $\ell_2$ regression is the Johnson Lindenstrauss transform (JLT) (Johnson & Lindenstrauss, 1984; Sarlos, 2006). In Sheffet (2017), it was shown that JLT itself preserves differential privacy when the data matrix satisfies certain conditions. Sheffet (2019) also noted that adding noise to the data matrix for the purpose of private linear regression results in a ridge regression problem where the regularization coefficient can be set such that the solution is differentially private. Note that this is unlike the standard regularized regression problems, where the regularization coefficient is set in order to the minimize the risk. In this work, we show that when noise is added in order to make the sketches private, solving the regression problem on the released private sketch is same as solving a sketched regularized regression problem (for both $p = 1$ and $p = 2$). We derive bounds for the regularization coefficient in such cases. However, when using the JLT for releasing private sketches, we show that the noise addition method can be altered such that the regression problem on the sketch is actually a sketched version of an "unregularized" regression problem. When the sketching matrix is `CountSketch` or the sketching algorithm of Munteanu et al. (2023), we still have to deal with a regularized regression problem.

**Organization of the paper** In Section 2, we introduce some definitions as well as the notations to be used in the paper. Section 3 proposes private sketches for the $\ell_2$ regression problem while Section 4 proposes sketches for the $\ell_1$ regression problem.

## 2 Preliminaries

We consider datasets of the form $(X, y)$ where $X \in \mathbb{R}^{n \times d}$ is the data matrix and $y \in \mathbb{R}^n$ is the response vector. In most of the discussions, we take $A = [X; y]$, where $y$ is appended to the columns of $X$ resulting in the $n \times (d+1)$ matrix $A$. Two datasets $A, A'$ are *neighbouring* if they differ in a single row. We assume that the $\ell_2$ norm of the rows of $A$ are bounded by $B$. Also, the matrix $A$ is assumed to have full column rank.

Let $A = U\Sigma V^\top$ be the singular value decomposition (SVD) of $A$. We use the notation $\sigma_{min}(A)$ to denote the minimum singular value of $A$. The $\ell_p$ regression loss is denoted by $||A\beta_{-1}||_p^q = ||X\beta - y||_p^q$, where $\beta_{-1} = \begin{bmatrix} \beta \\ -1 \end{bmatrix}, \beta \in \mathbb{R}^d$. The symbols $\mu, \nu$ denotes approximation error and failure probability respectively, such that $0 < \mu, \nu < 1$. We denote the indicator function by $\mathbb{1}$. The absolute value of a quantity is denoted by $|.|$. The multivariate isotropic normal distribution with mean 0 and variance $\sigma^2$ will be denoted by $\mathcal{N}(0, \sigma^2 I_d)$ where $I_d$ is the $d \times d$ identity matrix.

We define some of the concepts that will be used in the paper.

**Definition 1** (Differential Privacy Dwork et al. (2006)). *A randomized algorithm $\mathcal{M}$ is $(\epsilon, \delta)$-differentially private $((\epsilon, \delta) - DP)$ if for all neighbouring datasets $A, A'$ and for all possible outputs $\mathcal{E}$ in the range of $\mathcal{M}$,*

$$Pr(\mathcal{M}(A') \in \mathcal{E}) \leq \exp(\epsilon) \cdot Pr(\mathcal{M}(A) \in \mathcal{E}) + \delta. \tag{1}$$

**Definition 2** ($\ell_2$ sensitivity). *The $\ell_2$ sensitivity of a function $f : \mathbb{R}^{n \times d} \to \mathbb{R}^{r \times d}$ is defined as*

$$\Delta(f) = \sup_{neighbours \ A, A'} ||f(A) - f(A')||_2.$$

**Definition 3** (Gaussian Mechanism Dwork et al. (2006))**.** *The Gaussian mechanism $\mathcal{M}$ with noise level $\sigma = \frac{\Delta(f)}{\epsilon}\sqrt{2\ln(1.25/\delta)}$ is defined as*

$$\mathcal{M}(A) = f(A) + \mathcal{N}(0, \sigma^2 I_d).$$

The Gaussian mechanism is $(\epsilon, \delta)$- differentially private.

**Definition 4** (The Johnson-Lindenstrauss Lemma Johnson & Lindenstrauss (1984); Sarlos (2006))**.** *Let $0 < \mu, \nu < 1$ and $S = \frac{1}{\sqrt{r}} R \in \mathbb{R}^{r \times n}$ such that $R_{ij} \sim \mathcal{N}(0, 1)$ are independent random variables. If $r = \Omega(\frac{\log d/\nu}{\mu^2})$, then for a d-element subset $V \subset \mathbb{R}^n$, with probability at least $1 - \nu$, we have*

$$(1 - \mu)||v||_2^2 \leq ||Sv||_2^2 \leq (1 + \mu)||v||_2^2, \tag{2}$$

*for all $v \in V$. The matrix $S$ is the JL Transform.*

**Definition 5** (Subspace Embedding Clarkson & Woodruff (2017))**.** *A $poly(d, \mu^{-1}) \times n$ matrix $S$ is said to be a subspace embedding for $\ell_2$ for a fixed $n \times d$ matrix $X$, if for all $v \in \mathbb{R}^d$,*

$$(1 - \mu)||Xv||_2 \leq ||SXv||_2 \leq (1 + \mu)||Xv||_2,$$

*with probability at least* 9/10.

Sohler & Woodruff (2011); Woodruff & Zhang (2014) showed that subspace embeddings for the $\ell_1$ norm have $O(d \log d)$ dilation. The *sketch and solve* paradigm for linear regression Woodruff (2014) consists of two steps :

1. Compute a sketch , $SX$, of the data matrix $X$, where $S$ is the sketching matrix. The number of rows in $SX$ is less than in $X$.

2. Compute $\beta' = \underset{\tilde{\beta}}{\text{argmin}} ||SX\tilde{\beta} - Sy||_p^q$ on the smaller sketch.

Previous works have shown that the solution obtained on the sketched data is a good approximation to the original problem. For $\ell_2$ regression, it was shown in Sarlos (2006); Clarkson & Woodruff (2017) that $||X\beta' - y||_2 \leq (1 + \mu)||X\beta^* - y||_2$ with high probability. Also, for the case of $\ell_1$ regression, Munteanu et al. (2023) constructed a sketching matrix $S$ which was able to obtain an $O(1)$ approximation to the original problem.

## 3 Private Sketching for $\ell_2$ regression

The Johnson Lindenstrauss transform (JLT) Johnson & Lindenstrauss (1984) has been shown to preserve $(\epsilon, \delta)$-differential privacy when used for publishing a sanitized covariance matrix Blocki et al. (2012) of the data matrix $A$. A necessary condition for the privacy to be preserved is that $\sigma_{min}(A)$ must be sufficiently large. Following this, Sheffet (2017; 2019) used the JLT for computing a private JL projection of $A$. This private sketch was then used for computing the solution to the $\ell_2$ regression problem.

The method proposed by Sheffet (2017) required that $\sigma_{min}(A) \geq w$, for a threshold $w$, such that the algorithm for computing the sketches is differentially private. In case $\sigma_{min}(A) < w$, a matrix $w.I_{d+1}$ is appended to the rows of $A$ to get $\hat{A} = \begin{bmatrix} A \\ w.I_{d+1} \end{bmatrix} \in \mathbb{R}^{(n+d+1) \times (d+1)}$ such that the $\sigma_{min}(\hat{A}) \geq w$. The effect of concatenating $wI_d$ is to shift the singular values by $w$ such that $\sigma_{min}(\hat{A})$ becomes greater than or equal to $w$ (Blocki et al., 2012).

$$\begin{aligned}
\hat{A}^\top \hat{A} &= A^\top A + w^2 I_{d+1} = A^\top A + w^2 V V^\top \\
&= V(\Sigma^2 + w^2 I_{d+1})V^\top \\
&= V\sqrt{\Sigma^2 + w^2 I_{d+1}} U^\top U \sqrt{\Sigma^2 + w^2 I_{d+1}} V^\top.
\end{aligned} \tag{3}$$

When $\sigma_{min}(A) \geq w$, let $S \in \mathbb{R}^{r \times n}$ be the JLT (for a suitably chosen $r$). The private sketch released in this case is $SA$. Let $\beta = \underset{\tilde{\beta} \in \mathbb{R}^d}{\text{argmin}} ||SX\tilde{\beta} - Sy||_2^2$ Then, for $r = \Omega(\mu^{-2} d \log d)$, using the guarantees of JLT and from Theorem 12 of Sarlos (2006), with probability at least 1/3, we have

$$||SA\beta_{-1}||_2^2 \leq (1+\mu)||A\beta_{-1}||_2^2 = (1+\mu)||X\beta - y||_2^2 \leq (1+\mu)^3||X\beta^* - y||_2^2. \tag{4}$$

Here, since no alterations have to be made in order to guarantee privacy, solving the regression problem on $SA$ is same as solving the problem on a sketch of the unregularized problem.

In case $\sigma_{min}(A) < w$, let $S = [S_1; S_2]$ such that $S_1 \in \mathbb{R}^{r \times n}$ and $S_2 \in \mathbb{R}^{r \times (d+1)}$, for $r = \Omega(\mu^{-2} d \log d)$. The private sketch released is $S\hat{A}$. The least squares problem now becomes $\underset{\tilde{\beta}_{-1}}{\min} ||S\hat{A}\tilde{\beta}_{-1}||_2^2$. As noted in Sheffet (2019), $||\hat{A}\tilde{\beta}_{-1}||_2^2 = ||A\tilde{\beta}_{-1}||_2^2 + w^2||\tilde{\beta}_{-1}||_2^2$, i.e, it is an instance of the ridge regression problem where the parameter $w^2$ is set to preserve $(\epsilon, \delta)$-differential privacy.

Then, we get $||S\hat{A}\tilde{\beta}_{-1}||_2^2 \leq (1+\mu)||\hat{A}\tilde{\beta}_{-1}||_2^2 = (1+\mu)\left(||A\tilde{\beta}_{-1}||_2^2 + w^2||\tilde{\beta}_{-1}||_2^2\right)$, by the JL guarantee. From Sarlos (2006) and the JL guarantee, for $\beta = \underset{\tilde{\beta} \in \mathbb{R}^d}{\text{argmin}} ||S_1 X\tilde{\beta} - S_1 y||_2^2$, we have

$$\begin{aligned}
||S\hat{A}\beta_{-1}||_2^2 &\leq (1+\mu)||\hat{A}\beta_{-1}||_2^2 \\
&\leq (1+\mu)^3 ||\hat{A}\beta_{-1}^*||_2^2 \\
&\leq (1+\mu)^3 \left(||A\beta_{-1}^*||_2^2 + w^2||\beta_{-1}^*||_2\right) \\
&\leq (1+\mu)^3 \left(||X\beta^* - y||_2^2 + w^2||\beta_{-1}^*||_2^2\right)
\end{aligned} \tag{5}$$

with constant probability. In this case, since we have to append $w.I_{d+1}$ in order to guarantee privacy, solving the regression problem on $S\hat{A}$ is same as solving the problem on a sketch of a ridge regression problem where the regularization coefficient is $w^2 = \frac{8B^2}{\epsilon}\left(\sqrt{2r \ln(8/\delta)} + 2\ln(8/\delta)\right)$ (Sheffet, 2017).

### 3.1 Private JL Sketch with no regularization

Here, we show that when $\sigma_{min}(A) = \gamma.w$, for $0 < \gamma < 1$ and $\gamma$ is close to 1, then the sketching algorithm of Sheffet (2017) can be modified such that solving the regression problem on the private sketch does not require solving a regularized least squares problem. We want to form a matrix $\hat{A}$ such that $\sigma_{min}(\hat{A}) \geq w$. Let $Q \in \mathbb{R}^{(d+1) \times (d+1)}$ be a matrix such that, for some scalar $c$, we append $cQ$ to get $\hat{A} = \begin{bmatrix} A \\ cQ \end{bmatrix}$. Similar to Equation 3, we want that concatenating $cQ$ increases the singular values of the resulting matrix $\hat{A}$. Thus, we have $\hat{A}^\top \hat{A} = A^\top A + c^2 Q^\top Q$. If $Q^\top Q = V\Sigma^2 V^\top$, then we get $\hat{A}^\top \hat{A} = V\Sigma^2(1+c^2)V^\top$. This shifts the singular values of $A$ by $c^2\Sigma^2$. Thus, we set $Q = V\Sigma V^\top$ and $c = \sqrt{\frac{w^2}{\sigma_{min}(A)^2} - 1} = \sqrt{\frac{1}{\gamma^2} - 1}$. Also, note that $Q^\top Q = V\Sigma^2 V^\top = A^\top A$.

Let $S = [S_1; S_2]$ where $S_1 \in \mathbb{R}^{r \times n}$ and $S_2 \in \mathbb{R}^{r \times (d+1)}$ for $r = \Omega(\mu^{-2} d \log d)$. The private sketch released in this case is $S\hat{A}$. The least-squares problem on the released sketch then becomes $\underset{\tilde{\beta}_{-1}}{\min} ||S\hat{A}\tilde{\beta}_{-1}||_2^2$. Let $\beta = \underset{\tilde{\beta}}{\text{argmin}} ||S_1 X\tilde{\beta} - S_1 y||_2^2$. Then, using the guarantees of the JLT and Theorem 12 of Sarlos (2006), with constant probability, we have

$$\begin{aligned}
||S\hat{A}\beta_{-1}||_2^2 &\leq (1+\mu)||\hat{A}\beta_{-1}||_2^2 \\
&\leq (1+\mu)^3 ||\hat{A}\beta_{-1}^*||_2^2 \\
&= (1+\mu)^3(||A\beta_{-1}^*||_2^2 + c^2||Q\beta_{-1}^*||_2^2) \\
&= (1+\mu)^3(1+c^2)||A\beta_{-1}^*||_2^2 \\
&= (1+\mu)^3(1+c^2)||X\beta^* - y||_2^2
\end{aligned} \tag{6}$$

where the equality in the fourth step comes from $||Q\beta^*_{-1}||^2_2 = \beta^{*\top}_{-1}Q^\top Q\beta^*_{-1} = \beta^{*\top}_{-1}A^\top A\beta^*_{-1} = ||A\beta^*_{-1}||^2_2$. Thus, by concatenating $A$ with $cQ$, the singular values are scaled up by a factor of $1 + c^2 = 1/\gamma^2$. Solving the regression problem on the private sketch is same as solving an unregularized regression problem.

Algorithm 1 presents the proposed method. The values of the input parameters are same as in Sheffet (2017). The proof of privacy of Algorithm 1 is same as in (Sheffet, 2017).

---

**Algorithm 1** Private JL Sketch

---

**Require:** A matrix $A \in \mathbb{R}^{n \times (d+1)}$ and a bound $B$ on the $\ell_2$-norm of any row of $A$,
1: privacy parameters: $\epsilon, \delta > 0$
2: parameter $r$ denoting number of rows in the projected matrix
3: Set $w$ s.t. $w^2 = \frac{8B^2}{\epsilon}\left(\sqrt{2r\ln(8/\delta)} + 2\ln(8/\delta)\right)$
4: Sample $Z \sim Lap(4B^2/\epsilon)$ and let $\sigma_{min}(A)$ denote the smallest singular value of $A$.
5: **if** $\sigma_{min}(A)^2 > w^2 + Z + \frac{4B^2\ln(1/\delta)}{\epsilon}$ **then**
6:      Sample a $(r \times n)$ matrix $\hat{S}$ whose entries are i.i.d sampled from $\mathcal{N}(0,1)$
7:      **return** $SA$
8: **else**
9:      $Q = V\Sigma V^\top$, where $A = U\Sigma V^\top$
10:      $c = \sqrt{\frac{w^2}{\sigma_{min}(A)^2} - 1}$
11:      Let $\hat{A} = \begin{bmatrix} A \\ cQ \end{bmatrix}$
12:      Sample a $(r \times (n + d + 1))$ matrix $S$ whose entries are sampled i.i.d from $\mathcal{N}(0,1)$
13:      **return** $S\hat{A}$
14: **end if**

---

### 3.2 Private CountSketch for $\ell_2$ regression

Here, we look at another type of sketching method, using the `CountSketch` matrix, and discuss the alterations such that the sketching algorithm becomes differentially private. The CountSketch matrix gives low-distortion subspace embeddings for the $\ell_2$ norm in a data oblivious manner, and has been shown to give $(1 + \mu)$-approximate solution to the $\ell_2$ regression problem (Clarkson & Woodruff, 2017; Nelson & Nguyên, 2013; Meng & Mahoney, 2013). In Zhao et al. (2022), it was shown that a noisy initialization (calibrated to the $\ell_2$ sensitivity) results in a differentially private count sketch for the purpose of frequency estimation. Releasing private sketches for various downstream problems have also been proposed in (Li et al., 2019; Coleman & Shrivastava, 2021).

Let $S \in \{-1, 0, 1\}^{r \times n}$ be a count sketch matrix with $r \ll n$. Each column of $S$ has a single non-zero entry the coordinate for which is chosen uniformly at random. Each non-zero entry is either 1 with probability $1/2$ or $-1$ with probability $1/2$. The $i$th row of the sketch, $(SA)_{i,:}$ sums up the rows of $A$ (multiplied by $\pm 1$) that are present at the indexes corresponding to the non-zeros of $S_{i,:}$, and every row of $A$ is chosen exactly once. Thus, $(SA)_{i,:} = \sum_{j:\mathbb{1}\{S_{ij} \in \{-1,1\}\}} (\pm 1).A_{j,:}$. Let $A'$ be a neighbouring dataset that does not agree with $A$ on the $k$th row. Then, the $\ell_2$ sensitivity of computing $SA$ can be bounded as

$$\Delta = ||SA - SA'||_2 = ||(SA)_k - (SA')_{k,:}||_2 = ||A_{k,:} - A'_{k,:}||_2 \leq 2B. \tag{7}$$

So, if we add a noise vector $\tilde{\eta} \sim \mathcal{N}(0, \frac{8B^2\ln(1.25/\delta)}{\epsilon^2}I_{d+1})$ to each row of $SA$, then the algorithm releases $SA + \eta$ as the private sketch and it is $(\epsilon, \delta)$-differentially private owing to the Gaussian mechanism. Here, $\eta$ is a $r \times (d+1)$ matrix where each row is a noise vector $\tilde{\eta}_i, i = 1, \ldots, r$. Then, the $\ell_2$ regression problem becomes $\min_{\beta_{-1}} ||(SA + \eta)\beta||^2_2$. But we cannot use Theorem 30 of Clarkson & Woodruff (2017) to give approximation guarantees for $||SA\beta_{-1}||^2_2$. Hence, we introduce the Gaussian noise for preserving privacy differently.

Let $\eta$ be a $p \times (d+1)$ matrix such that each row of $\eta$ is sampled independently from $\mathcal{N}(0, \frac{8B^2 \ln(1.25/\delta)}{\epsilon^2} I_{d+1})$. We concatenate $\eta$ to the rows of $A$ in order to get $\hat{A} = \begin{bmatrix} A \\ \eta \end{bmatrix}$ which is a $(n+p) \times (d+1)$ matrix. Let the $r \times (n+p)$ countsketch matrix be $S = [S_1; S_2]$ where $S_1 \in \{-1, 0, 1\}^{r \times n}$ and $S_2 \in \{-1, 0, 1\}^{r \times p}$ are also countsketch matrices. Then, the private sketch to be released is $S\hat{A}$. The effect of $S_2\eta$ is to add noise vectors to each of the $r$ rows of $S_1 A$. Now, we specify the number of rows $p$ of $\eta$.

In order to make the sketch $S\hat{A}$ private, we want to add the noise vectors from $\eta$ to each of the $r$ rows of $S_1 A$. Thus, we want that $S_2$ maps the rows of $\eta$ to each row of $S\hat{A}$ at least once. This is analogous to the Coupon Collector's problem in that we have to collect $r$ coupons at least once in order to win. The expected number of trials required is then $p = O(r \log r)$. Thus, the noise matrix $\eta$ is a $O(r \log r) \times (d+1)$ matrix.

Let $\beta = \underset{\tilde{\beta} \in \mathbb{R}^d}{\text{argmin}} ||SX\tilde{\beta} - Sy||_2^2$. Then, from the subspace embedding guarantees of the CountSketch matrix as well as $\ell_2$ regression approximation guarantees from Meng & Mahoney (2013), with constant probability we have

$$
\begin{aligned}
||S\hat{A}\beta_{-1}||_2^2 &\le (1+\mu)||\hat{A}\beta_{-1}||_2^2 \\
&\le (1+\mu)^3 ||\hat{A}\beta_{-1}^*||_2^2 \\
&= (1+\mu)^3 \left( ||A\beta_{-1}^*||_2^2 + ||\eta\beta_{-1}^*||_2^2 \right) \\
&= (1+\mu)^3 \left( ||X\beta^* - y||_2^2 + ||\eta\beta_{-1}^*||_2^2 \right).
\end{aligned}
\tag{8}
$$

From the inequality at 8, we observer that solving the least squares regression problem on the private sketch $S\hat{A}$ results in solving a sketched ridge regression problem. In Theorem 1, we bound the regularization coefficient.

**Theorem 1.** *Let $S \in \{-1, 0, 1\}^{r \times (n+r \log r)}$ be the sketching matrix which has been described above and let $r = poly(d, \mu^{-1})$. Also, let $\eta$ be a $r \log r \times (d+1)$ matrix where the ith row $\tilde{\eta}_i \sim \mathcal{N}(0, \frac{8B^2 \ln(1.25/\delta)}{\epsilon^2} I_{d+1})$. Then, solving the $\ell_2$ regression problem on $S\hat{A}$ is same as solving a sketched ridge regression problem where the regularization coefficient is*

$$
||\eta\beta_{-1}||_2 \le \frac{13B}{\epsilon} \sqrt{r \log r \ln(1.25/\delta)} ||\beta_{-1}||_2
\tag{9}
$$

*with constant probability.*

*Proof.* Let $\sigma = \frac{2B}{\epsilon} \sqrt{2\ln(1.25/\delta)}$. Each $\tilde{\eta}_i$ is sampled from $\mathcal{N}(0, \sigma^2 I_{d+1})$. So, we have $\tilde{\eta}_i^\top \beta_{-1} \sim \mathcal{N}(0, \sigma^2 ||\beta_{-1}||_2^2)$. We are interested in bounding $||\eta\beta_{-1}||_2 = \left( \sum_{i=1}^{r \log r} (\tilde{\eta}_i^\top \beta_{-1})^2 \right)^{\frac{1}{2}}$, where $\eta\beta_{-1}$ is an $(r \log r)$-dimensional vector of i.i.d normal random variables. From Ledoux & Talagrand (1991), for $t \ge 0$, we have

$$
Pr(||\eta\beta_{-1}||_2 \ge t) \le 4 \exp\left( -\frac{t^2}{8\mathbb{E}[||\eta\beta_{-1}||_2^2]} \right).
\tag{10}
$$

Here, $\mathbb{E}[||\eta\beta_{-1}||_2^2] = \sum_{i=1}^{r \log r} \mathbb{E}[(\tilde{\eta}_i^\top \beta_{-1})^2] = r \log(r) \sigma^2 ||\beta_{-1}||_2^2$. Suppose $||\eta\beta_{-1}||_2 \ge t$ with probability at most $1/4$. Then,

$$
\begin{aligned}
&4 \exp\left( -\frac{t^2}{8\mathbb{E}[||\eta\beta_{-1}||_2^2]} \right) \le \frac{1}{4} \\
\implies &\frac{t^2}{8\mathbb{E}[||\eta\beta_{-1}||_2^2]} \ge \ln(16) \\
\implies &t \ge \sqrt{8\mathbb{E}[||\eta\beta_{-1}||_2^2]\ln(16)} \approx \frac{13B}{\epsilon} \sqrt{r \log(r) \ln(1.25/\delta)} ||\beta_{-1}||_2.
\end{aligned}
\tag{11}
$$

$\square$

## 4   Private Sketching for $\ell_1$ regression

An analogue of the JLT for the problem of $\ell_1$-norm is the Cauchy transform (Sohler & Woodruff, 2011; Clarkson et al., 2016), though it has weaker guarantees (lopsided subspace embedding) . The Cauchy transform is used for speeding up the computation of a distribution that is used for sampling from the rows of $A$. This algorithm was proposed in Clarkson (2005) to obtain a $(1 + \mu)$ approximation to the solution of the $\ell_1$ regression problem. However, this algorithm outputs a weighted subsample from the dataset itself, which could lead to the leakage of private information.

Instead we look at a recently proposed oblivious linear sketching technique that satisfies the conditions of a lopsided embedding and gives an $O(1)$ approximation to the solution of the $\ell_1$ regression problem (Munteanu et al., 2023). Similar sketches have also been utilized in (Clarkson & Woodruff, 2015; Munteanu et al., 2021). The sketch is analogous to a concatenation of multiple `CountMinSketches.` Similar to Section 3.2, we append a noise matrix to the rows of $A$ and release a sketch of this augmented matrix as the private sketch. We use the sketching algorithm of Munteanu et al. (2023).

### 4.1   An Illustration

For the purpose of illustration, we consider a simpler sketching matrix $S \in \{0, 1\}^{r \times n}$ where each column of $S$ has a single entry set to 1. Also, for any vector $\beta_{-1}$, assume that $||SA\beta||_1 \leq O(1)||A\beta||_1$, with constant probability. The $i$th row of $SA$ can be written as $(SA)_{i,:} = \sum_{j:\mathbb{1}\{S_{ij}=1\}} A_{j,:}$. Let $A'$ be the neighbouring dataset such that $A$ and $A'$ differ in the $k$th row. Then, the $\ell_2$ sensitivity of $SA$ can be bounded as

$$\Delta = ||SA - SA'||_2 = ||(SA)_{k,:} - (SA')_{k,:}||_2 = ||A_{k,:} - A'_{k,:}||_2 \leq ||A_{k,:}||_2 + ||A'_{k,:}||_2 \leq 2B. \tag{12}$$

Thus, if we add a noise vector $\tilde{\eta} \sim \mathcal{N}(0, \frac{8B^2 \ln(1.25/\delta)}{\epsilon^2} I_{d+1})$, then the algorithm that samples $S$ and releases $SA$, is $(\epsilon, \delta)$-DP, by the Gaussian mechanism. But, as discussed in Section 3.2, we will not be able to give approximation guarantees on the sketched $\ell_1$ problem , $||(SA+\eta)\beta_{-1}||_1$. Here, $\eta$ is a $r \times (d+1)$ noise matrix for which each row is $\tilde{\eta}_i, i = 1, \ldots, r$.

Instead we append a $p \times (d + 1)$ noise matrix $\eta$ to the rows of $A$ to get $\hat{A} = \begin{bmatrix} A \\ \eta \end{bmatrix}$ and release the sketched matrix $S\hat{A}$ as the private sketch. Using the Coupon Collector's argument as before, $\eta$ is a $O(r \log r) \times (d+1)$ matrix. Solving the $\ell_1$ regression problem on $S\hat{A}$ is same as solving a sketched regularized $\ell_1$ regression problem. Let $\beta = \underset{\tilde{\beta}}{\mathrm{argmin}} ||SX\tilde{\beta} - Sy||_1$, where $S \in \{0, 1\}^{r \times (n+r \log r)}$ (Gordon et al., 2006). Then, with constant probability, we have

$$\begin{aligned} ||S\hat{A}\beta_{-1}||_1 &\leq O(1)||\hat{A}\beta_{-1}||_1 \\ &\leq O(1)||\hat{A}\beta^*_{-1}||_1 \\ &= O(1)\left(||A\beta^*_{-1}||_1 + ||\eta\beta^*_{-1}||_1\right) \\ &= O(1)\left(||X\beta^* - y||_1 + ||\eta\beta^*_{-1}||_1\right). \end{aligned} \tag{13}$$

We will be using Lemma 1 for deriving a bound on the regularization coefficient.

**Lemma 1.** *Let $U = [u_1, \ldots, u_r]$ be a vector of independent and identically distributed (i.i.d) random variables with $u_i \sim \mathcal{N}(0, \sigma^2)$. Let $||U||_1 = \sum_{i=1}^{r} |u_i| = S_r$. Then, with probability at least $\frac{3}{4}$,*

$$||U||_1 \leq r\sigma \tag{14}$$

*Proof.* Since $u_i \sim \mathcal{N}(0, \sigma^2)$, the random variable $Q = |u_i|$ has a half normal distribution. For $t \in \mathbb{R}$, the moment generating function (MGF) of $Q$ is

$$M_Q(t) = \mathbb{E}[\exp(t|u_i|)] = \exp(\frac{\sigma^2 t^2}{2})\mathrm{erfc}(-\frac{\sigma t}{\sqrt{2}}) = \exp(\frac{\sigma^2 t^2}{2})(1 + \mathrm{erf}(\frac{\sigma t}{\sqrt{2}})), \tag{15}$$

using the property that $\operatorname{erf}(-z) = -\operatorname{erf}(z), \operatorname{erf}(.)$ being the Gaussian error function. Since, $u_i$s are i.i.d random variables, so too are the $|u_i|$s. Then, for $t \in \mathbb{R}$, the MGF of $S_r$ becomes $M_{S_r}(t) = \mathbb{E}[\exp(tS_r)] = \mathbb{E}[\exp(t\sum_{i=1}^{r} |u_i|)] = \prod_{i=1}^{r} \mathbb{E}[t\sum_{i=1}^{r} |u_i|)] = [M_Q(t)]^r$. Applying Chernoff bound on the sum of i.i.d random variables, we have

$$
\begin{aligned}
Pr(S_r \geq a) &\leq \inf_{t>0} \exp(-ta)\prod_{i=1}^{r}\mathbb{E}[\exp(t|u_i|)] \\
&= \inf_{t>0} \exp(-ta)[\exp(\frac{\sigma^2 t^2}{2})(1 + \operatorname{erf}(\frac{\sigma t}{\sqrt{2}}))]^r
\end{aligned}
\tag{16}
$$

Let $g(t) = \exp(-ta)[\exp(\frac{\sigma^2 t^2}{2})(1 + \operatorname{erf}(\frac{\sigma t}{\sqrt{2}}))]^r$. Then, we get

$$
\begin{aligned}
\ln(g(t)) &= \ln(\exp(-ta)[\exp(\frac{\sigma^2 t^2}{2})(1 + \operatorname{erf}(\frac{\sigma t}{\sqrt{2}}))]^r) \\
&= -ta + \frac{r\sigma^2 t^2}{2} + r\ln(1 + \operatorname{erf}(\frac{\sigma t}{\sqrt{2}})) \\
&< -ta + \frac{r\sigma^2 t^2}{2} + r\ln 2,
\end{aligned}
\tag{17}
$$

since $\operatorname{erf}(.) \in (-1, 1)$. The minimizer of $\ln(g(t))$ is at $t' = \frac{a}{r\sigma^2}$. At the minimizer, we get $\ln(g(t')) = \ln(2^r) - \frac{a^2}{2r\sigma^2}$ which gives $g(t') = \exp(ln(2^r))\exp(-\frac{a^2}{2r\sigma^2}) = 2^r \exp(-\frac{a^2}{2r\sigma^2})$. Thus,

$$
Pr(S_r \geq a) \leq 2^r \exp(-\frac{a^2}{2r\sigma^2}).
\tag{18}
$$

If $Pr(S_r \geq a) \leq \frac{1}{4}$, then

$$
\begin{aligned}
2^r \exp(-\frac{a^2}{2r\sigma^2}) &\leq \frac{1}{4} \\
\implies a &\geq r\sigma.
\end{aligned}
\tag{19}
$$

$\square$

This leads us to Lemma 2.

**Lemma 2.** *With constant probability,*

$$
||\eta\beta_{-1}||_1 \leq \frac{2Br \log r \sqrt{2\ln(1.25/\delta)}}{\epsilon}||\beta_{-1}||_1.
\tag{20}
$$

*Proof.* We have $||\eta\beta_{-1}||_1 = \sum_{i=1}^{r \log r} |\eta_i\beta_{-1}|$, where $\eta_i$ is the $i$th row of $\eta$. Also, each coordinate of $\eta_i$ is sampled i.i.d from $\mathcal{N}(0, \frac{8B^2 \ln(1.25/\delta)}{\epsilon^2})$. Thus, $\eta_i\beta_{-1} \sim \mathcal{N}(0, \frac{8B^2 \ln(1.25/\delta)}{\epsilon^2}||\beta_{-1}||_2^2)$. Therefore, $||\eta\beta_{-1}||_1$ is the sum of $r \log r$ i.i.d random variables each sampled from $\mathcal{N}(0, \frac{8B^2 \ln(1.25/\delta)}{\epsilon^2}||\beta_{-1}||_2^2)$. Hence, by Lemma 1, we have

$$
||\eta\beta_{-1}||_1 \leq \frac{2Br \log r \sqrt{2\ln(1.25/\delta)}}{\epsilon}||\beta_{-1}||_2 \leq \frac{2Br \log r \sqrt{\ln(1.25/\delta)}}{\epsilon}||\beta_{-1}||_1,
\tag{21}
$$

with probability at least $\frac{3}{4}$. $\square$

Now, we consider the sketching algorithm of Munteanu et al. (2023) and apply the modifications to make it private.

### 4.2 Private $\ell_1$ Sketch

The sketching algorithm given in Munteanu et al. (2023) outputs a weak weighted sketch that obtains an $O(1)$ approximation to the $\ell_1$ regression problem. The sketching dimension is $r = O(d^{1+c} \ln(n)^{3+5c}), 0 < c \leq 1$ for constant success probability. The sketching algorithm of Munteanu et al. (2023) also outputs an oblivious weight vector $w$.

Since, we append a $r \log r \times (d+1)$ noise matrix to $A$ (to get the noise augmented matrix $\hat{A}$), the weight vector $w$ is also of length $r \log r$. Note that the weight vector as well as the sketch is data oblivious. We give a brief description of the sketching algorithm in Munteanu et al. (2023). The sketching matrix $S$ consists of $O(\log n)$ levels of sketch as shown.

$$S = \begin{bmatrix} S_0 \\ S_1 \\ \vdots \\ S_{h_m} \end{bmatrix}. \tag{22}$$

Here, $h_m = O(\log_b n)$ is the number of sketch levels, where $b \in \mathbb{R}$ is a branching parameter. The number of buckets(rows) at level $h$ is denoted by $N_h$. The probability that any row $A_{i,:}$ is sampled at level $h$ is $p_h \propto 1/b^h$. The rows of $A$ that are mapped to the same row of the sketch are added up. Also, the weight of any bucket at level $h$ is set to $1/p_h$.

The sampling probabilities decrease exponentially as the levels increase. At level 0, sketching with $S_0$ is a `CountMinSketch` of the entire dataset. It captures the elements that make a significant contribution to the objective, the so-called heavy hitters. For improving efficiency, the rows at level 0 are mapped to $s \geq 1$ rows of the sketch. On the other hand, the sketch at level $h_m$ corresponds to a small uniform subsample of the data. For each sketch level $S_i$, the probability of having a 1 at any of the columns is $1/b^i$. For a detailed analysis of the algorithm, readers may refer to (Munteanu et al., 2023).

The private sketch, $S\hat{A}$, sums up the corresponding rows of $A$ and at least one of the noisy rows from $\eta$. The details are given in Algorithm 2. It is similar to the algorithm of Munteanu et al. (2023) with a few alterations.

In Algorithm 2, a row can be sampled at most $h_m$ times. Therefore, the $\ell_2$ sensitivity of the applying the sketch is at most $2B\sqrt{h_m}$. We can now state Theorem 2.

**Theorem 2.** *Algorithm 2 satisfies $(\epsilon, \delta)$-differential privacy and outputs a weighted sketch $C = (S\hat{A}, w)$. Solving the $\ell_1$ regression problem on $S\hat{A}$ is same as solving a sketched regularized $\ell_1$ regression problem where the regularization coefficient is*

$$||\eta\beta_{-1}||_1 \leq \frac{2Br \log r \sqrt{2h_m \ln(1.25/\delta)}}{\epsilon} ||\beta_{-1}||_1. \tag{23}$$

*with constant probability.*

*Proof.* The differential privacy guarantee comes from the application of the Gaussian mechanism. The $\ell_2$ sensitivity of the sketch is $2B\sqrt{h_m}$. Applying Lemma 2, we get the bound on the regularization coefficient. □

## 5 Conclusion and Future Work

In this paper, we proposed a few techniques for releasing differentially private sketches of the dataset. We utilized the JL transform as well as sparse sketches for computing the sketches. We observed that in most of the cases, introducing noise in order to preserve privacy results in having to solve sketch regularized regression problems. The bounds on the regularization coefficients highlight the regularization effect for guaranteeing privacy. The coefficients could potentially be large, which could lead to a large drop in utility of the solutions on the private sketches. For the JL transform, we showed that it is possible to get unregularized regression

---

**Algorithm 2** DP oblivious sketching algorithm for $\ell_1$ regression.

---
**Input:** Data $A \in \mathbb{R}^{n \times (d+1)}$, number of rows $r = N \cdot h_m + N_u$, parameters $b > 1, s \geq 1$ where $N = s \cdot N'$ for some $N' \in \mathbb{N}$, privacy parameters $\epsilon, \delta$, $\ell_2$ row norm bound $B$;
**Output:** weighted Sketch $C = (SA, w) \in \mathbb{R}^{r \times d}$ with $r$ rows.;

1: $\sigma = \frac{2Bh_m}{\epsilon}\sqrt{2\ln(1.25/\delta)}$, where $h_m = O(\log_b n)$

2: $\hat{A} = \begin{bmatrix} A \\ \eta \end{bmatrix}$, where each row of $\eta$ is $\eta_i \sim \mathcal{N}(0, \sigma^2 I_{d+1})$.

3: **for** $h = 0 \ldots h_m$ **do**               ▷ construct levels $0, \ldots h_m$ of the sketch
4:      initialize sketch $S\hat{A}_h = 0$ at level $h$ ;
5:      initialize weights $w_h = b^h \cdot \mathbf{1} \in \mathbb{R}^N$ at level $h$;
6: **end for**
7: set $w_0 = \frac{w_0}{s}$;                          ▷ adapt weights on level 0 to sparsity $s$
8: **for** $i = 1 \ldots n$ **do**                       ▷ sketch the data
9:      **for** $l = 1 \ldots s$ **do**                 ▷ densify level 0
10:          draw a random number $B_i \in [N']$;
11:          add $\hat{A}_{i,:}$ to the $((l-1) \cdot N' + B_i)$-th row of $SA_0$;
12:      **end for**
13:      assign $\hat{A}_{i,:}$ to level $h \in [1, h_m - 1]$ with probability $p_h = \frac{1}{b^h}$;
14:      draw a random number $B_i \in [N]$;
15:      add $\hat{A}_{i,:}$ to the $B_i$-th row of $S\hat{A}_h$;
16:      add $S\hat{A}_{i,:}$ to uniform sampling level $h_m$ with probability $p_{h_m} = \frac{1}{b^{h_m}}$;
17: **end for**
18: Set $S\hat{A} = (S\hat{A}_0, S\hat{A}_1, \ldots S\hat{A}_{h_m})$;
19: Set $w = (w_0, w_1, \ldots w_{h_m})$;
20: **return** $C = (S\hat{A}, w)$;

---

formulations in certain cases. While for the $\ell_2$ regression, Algorithm 1 works, we explored the feasibility of releasing private CountSketch as well. A future direction of work is to come up with similar constructions for the other problems, especially $\ell_1$ regression, as well. Specifically, we would like to have a better noise addition mechanism such that the regularization coefficient does not become large.

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
