# OpenReview forum: "Private Sketches for Linear Regression"
_TMLR — Rejected by TMLR_

### Review · Reviewer_56ZW · 2026-02-01

**Summary Of Contributions:**

The authors propose a differentially private sketch-and-solve approach for linear regression. Rather than directly releasing the privitized sufficient statistics, they privately release sketches of the data matrix. The proposed framework applies to both L2 regression and L1 regression, via applying different sketching matrices.

**Additional Comments:**

The “sketch-then-solve” technique is not entirely new in the DP literature. Related ideas have been explored in the DP-ERM literature for private classification; see, for example, Algorithm 1 of Bassily, Mohri, and Suresh [1].

[1] Bassily, Raef, Mehryar Mohri, and Ananda Theertha Suresh. "Differentially private learning with margin guarantees." Advances in Neural Information Processing Systems 35 (2022): 32127-32141.

**Audience:**

No

**Audience Explanation:**

I would like to update my assessment after a more careful reading. The positioning of the paper is not entirely clear to me. If the paper is intended for a theory audience, then the methodological contribution may need further development. If it is aimed at practitioners, then a more comprehensive empirical comparison with more existing private regression methods is needed. I think the paper would be interested to TMLR audience after revisions.

**Claims And Evidence:**

No

**Claims Explanation:**

I would like to update my assessment after a more careful examination of the proof. While I initially found the main argument convincing, I now believe there is a gap in the analysis, as pointed out by other reviewers. Even after the rebuttal, I think some statements would still benefit from further clarification, and parts of the proof could be better organized. Rephrasing and reorganizing the relevant sections would greatly improve the readability of the paper.

**Requested Changes:**

(1) The paper adopts a largely narrative exposition, rather than a conventional theorem–algorithm–proof style (especially section 3 and 4.1). As a result, some of the technical guarantees and assumptions are not immediately apparent, and readers may need to reconstruct formal statements from the surrounding discussion. Reorganizing the presentation to more clearly separate definitions, formal results, and proofs would improve readability for the TMLR audience.

(2) Since the paper focuses on private linear regression, from a practical point of view, can the author provide experimental results to compare the proposed methods with existing baselines, such as Sheffet (2017), and demonstrate the privacy-utility trade-off?

---

> ### Author Response · Authors · 2026-02-05
> **Response to Reviewer 56ZW**
>
> We thank the reviewer for reading our paper and providing helpful reviews.
>
> **Requested Changes 1 :** We are working on restructuring the presentation to  reflect the changes suggested by the reviewer.
>
> **Requested Changes 2 :** Please refer to the official comments titled "Modifications to the Submission" and "Empirical Results".

---

### Review · Reviewer_Dd5L · 2026-03-03

**Summary Of Contributions:**

This paper advocates a private sketch paradigm for linear regression, instead of privately outputting a regression solution (or noisy sufficient statistics), the algorithm releases a differentially private sketch of the data. The paper studies several sketching instantiations, including JL-style transforms and CountSketch-type constructions, and interprets the privacy mechanism as inducing an implicit regularization effect in some cases.

**Audience:**

Yes

**Audience Explanation:**

Releasing a reusable DP sketch is an attractive alternative to one-off DP regression outputs, and it aligns well with the sketch-and-solve workflow widely used in numerical linear algebra and large-scale regression.

**Broader Impact Concerns:**

This work is positioned as a privacy-enhancing method that could enable safer sharing of regression data and repeated downstream analyses via post-processing. A potential concern is misuse or overclaiming: releasing a sketch with poorly chosen $(\epsilon,\delta)$ or unclear utility guarantees (especially if only constant-probability bounds are provided) may give practitioners a false sense of privacy or reliability. It would be helpful if the paper included clearer guidance on parameter selection and failure-probability/robustness in practical deployments.

**Claims And Evidence:**

No

**Claims Explanation:**

The paper makes a strong algorithmic and conceptual proposal, but it provides essentially no empirical evaluation. As a result, it is difficult to assess whether the proposed private sketches are competitive in practice, or whether the induced regularization/noise is so large that utility collapses in realistic regimes.

**Requested Changes:**

1. Can the main theoretical guarantees be strengthened from constant probability to $1-\nu$ with explicit dependence on $\nu$? If yes, what is the resulting impact on (i) the required sketch dimension, (ii) the noise magnitude / regularization strength, and (iii) the overall runtime? I would suggest the authors provide high-probability versions of the main theorems and clarify whether standard probability amplification techniques (e.g., repetition + median/selection) apply without changing the privacy accounting (or explain how privacy would change if multiple sketches are released).

2. I would suggest the authors add a dedicated experimental section with clear baselines.

---

> ### Author Response · Authors · 2026-04-15
> **Response to Reviewer Dd5L**
>
> We thank the reviewer for the  reviews that have helped us to improve the paper.
>
> **Regarding Modifications and Experiments**
> Please refer to the official comments titled "Modifications to the Submission" and "Empirical Results" .
>
> **Regarding Strengthening the Guarantees**
> This is an interesting suggestion. When we apply Countsketch directly, we would require repetition plus median selection type technique for probability amplification.  However, in our modified private countsketch, we precondition the data matrix with HD which has the affect of "spreading out" the mass of the vector. Our sketch now behaves similar to the Fast JL sketch, and probability of success can be amplified by increasing the size of the sketch.

---

### Review · Reviewer_Wu7J · 2026-04-04

**Summary Of Contributions:**

- This paper constructs differentially private sketches for both $\ell_2$ and $\ell_1$ regression.
- For $\ell_2$ regression, the authors show that for $\sigma_{\text{min}} (A)$ less than, but close to $w$, solving the regression problem on the private sketch does not require solving a regularized least squares problem. The authors also propose a CountSketch-based sketching, that involves explicit noise addition.
- I'm not listing the $\ell_1$ regression results here, since I have doubts about their correctness.

**Additional Comments:**

See above

**Audience:**

No

**Audience Explanation:**

Some of the results in the paper are problematic, for reasons described above. However, once they are corrected, I would like the authors to explain their significance, since some of them seem too straightforward. Therefore, I am a bit skeptical of the marginal utility of this paper. For example,

- Lemma 1 and Lemma 2, in their current form, are problematic. But, as I point out above, they can be derived in a straightforward way from standard Gaussian concentration results. Doesn't that make Theorem 2 straightforward as well, especially given the scheme Munteanu et al. (2023). The only contribution here is the application of coupon collector, right?
- In its current form, leaving the issue pointed out above, Theorem 1 is also quite straightforward. What is the novel insight used in the result?

**Claims And Evidence:**

No

**Claims Explanation:**

I have several questions and concerns about the theoretical arguments to support the results in the paper. It is possible (albeit unlikely) that I have misunderstood some things, so I look forward to the authors' responses.

### Issues in Theorem 1
- In the proof of Theorem 1, the matrix $S_2$ has $O(r \log r)$ rows. However, in this case, there is a significant probability that one of the rows of $S_1 A$ does not get any noise. Because $O(r \log r)$ is just an expectation, not a high-probability bound. How do we ensure that the probability of the failure case I just described does not break the privacy guarantee completely?
- Since the same methodology is used in Section 4.1, the same issue continues there as well.

### Issues in Lemma 1 and 2
- Lemma 1 does not seem to hold for $r=1$. So, I am not sure of the correctness of the result in its current form.
- Next, in the proof of Lemma 1, at the end, you say that if $Pr(S_r \leq a) \leq 1/4$, then (19) follows. Should it not be the other way round? If (19) follows, then $Pr(S_r \leq a) \leq 1/4$. Further, how does $a \geq r \sigma$ follow from the previous inequality in (19)? You should perhaps get something like $a \geq \sqrt{2 \ln 2} r \sigma$. Which brings us to the final point - what is the necessity of Lemma 1? We can get almost the same result with a high probability bound on $||U||_2$ norm along with the equivalence of $\ell_1, \ell_2$ norms.
- Consequently, the result of Lemma 2 does not hold either, in its current form.

**Requested Changes:**

- There is a typo in the MGF of $S_r$ in the paragraph above (16).
- After (17), is $t'=\frac{a}{r \sigma^2}$ the minimizer of $\ln(g(t))$ or of the upper bound? Also, what is given as $\ln(g(t'))$ is actually an upper bound, no?
- In Section 4.1, it is said that "for any vector $\beta−1$, assume that $\|SA\beta\|_1 \leq O(1)\|A\beta\|_1$, with constant probability." Isn't this a deterministic statement?

---

> ### Author Response · Authors · 2026-04-16
> **Response to Reviewer Wu7J**
>
> We thank the reviewer for the reviews and highlighting the inconsistencies.
>
> ### Issues in Theorem 1
> Please refer to official comment titled "Modifications to the submission". Apart from the issue raised by the reviewer that some rows of $S_1A$ may not get any noise, the earlier method also  destroyed the utility of the sketch. The new method alleviates these issues, as demonstrated in the empirical results as well.
>
> ### Issues in Lemma 1 and 2
> We acknowledge the errors in the current proof of Lemma 1 and consequently Lemma 2. We modify the statement for Lemma 1 and its proof. We present a revision of Lemma 1 here. The subsequent lemma statements will be updated accordingly.
>
> Lemma :
> Let $U = [u_1, \ldots, u_r]^\top$ be a vector of independent and identically distributed (i.i.d) random variables with $u_i \sim \mathcal{N}(0,\sigma^2).$ Let $||U||_1 = \sum\_{i=1}\^{r}|u\_i|.$ Then, with probability at least 3/4,$$||U||\_1 \leq r\sigma\sqrt{\frac{2}{\pi}} + 2\sigma\sqrt{r\ln(2)}$$
>
> Proof:
> Since $u_i \sim \mathcal N(0,\sigma^2),$ the random variable $|u_i|$ follows a folded normal distribution. Thus, $\mathbb{E}[|u_i|] = \sigma \sqrt{\frac{2}{\pi}}$ and by linearity of expectation, $\mathbb{E}[||U||_1] = r \sigma \sqrt{\frac{2}{\pi}}.$
>
> Using the fact that $f(U) = ||U||_1$ is $\sqrt{r}$-Lipschitz with respect to the Euclidean norm, we can apply concentration bounds on such functions of Gaussian randon variables [1] which states
> $$Pr\bigg(||U||_1 - r \sigma \sqrt{\frac{2}{\pi}} \geq t\bigg) \leq \exp\bigg(- \frac{t^2}{2 r \sigma^2}\bigg)$$ for any $t \geq 0.$
>
> Now, if $t \geq  2 \sigma \sqrt{r \ln 2},$ then
> $$Pr\bigg(||U||_1 \geq r \sigma \sqrt{\frac{2}{\pi}} +  2 \sigma \sqrt{r \ln 2} \bigg) \leq \frac{1}{4}.$$
>
> Proof End.
>
> ### Comment on the utility of the paper.
> We agree that the mathematical results of this paper are straightforward while the main contribution of this paper is a framework for releasing differentially private sketches. In the official comments, we show that we inject noise into the sketch using a  fast Johnson-Lindenstrauss inspired technique, which ensures that each row receives a noise addition. This is the first work aimed at releasing private sketches for regression. The privacy preserving techniques that we have proposed in the paper (and the official comments) are simple enough which enables their analysis using standard results. The simplicity of the methods also leads to their implementation being efficient. Based on reviewers' comments, we also demonstrated our method empirically. The empirical comparison of our private CountSketch and Sheffet's private JL sketch demonstrates the effectiveness of the approach in the small $\varepsilon$ regime, where usually the utility degrades rapidly (as is the case for Sheffet's sketch).
>
> ### Regarding Requested Changes
> 1. Since the lemma statement and the proof has changed, the typos do not need to be addressed.
> 2. We will rewrite the distortion for the $\ell_1$ sketch. Instead of $O(1)$ distortion, it must be a constant $C = \alpha/1-\mu,$ where $\alpha$ is the dilation bound for the sketch. It follows from the definition of the weak-weighted sketch given in Munteanu et al.
>
> We will update the revised manuscript with these changes.
>
> References:
> 1. High-dimensional Statistics : A Non-Asymptotic Viewpoint, Martin Wainwright, 2019.

---

### Author Response · Authors · 2026-04-15
**Modifications to the submission**

## Issues in Section 3.1
Due to issues with correctness in Section 3.1, we will be dropping that section from a revised version of the paper.

## Modifying the noise injection mechanism

Based on the reviews we have received, we experimented on a synthetic dataset using our proposed methods. For the private Countsketch and the private $\ell_1$ sketch, we observed that appending $r \log r$ rows of Gaussian noise degrades the utility of the sketches. This is because in both of the sketches $r$ depends polynomially on $d$ and inversely on the accuracy parameters. Increasing $r$ for improving utility results in adding more privacy noise, which in turn adversely affects the utility.

To mitigate this issue, we modify the noise injection mechanism in the following manner, inspired from the techniques in [1] . Let $H_{2^k}$ be the scaled Hadamard matrix of order $k$ (scaled by $1/\sqrt{2^k}$) and $D$  be a $2^k \times 2^k$ diagonal matrix whose entries are sampled from $\{-1,1\}$ with probability $1/2.$ Before applying the sketch $S,$ we compute $HD\hat{A},$ where $\hat{A} = \begin{bmatrix}
    A \\
    \eta
\end{bmatrix}$ is a $(n+p)\times(d+1)$ matrix. We set $k = \lceil{\log_2(n+p)}\rceil.$ If $n+p < 2^k,$ then we pad the remaining rows of $\hat{A}$ with zeros. Let the $r \times 2^k$ countsketch matrix be $S.$ Then, the private sketch to be released is $S(HD\hat{A}).$  The effect of $HD\hat{A}$ is to sum up the rows (with the sign flipped with probability $1/2$) of $\hat{A}$ which includes the rows of $A$ as well as the noisy rows. The amount of noisy rows added now becomes $p$ instead of $r \log r.$

We can show that the $\ell_2$ sensitivity of SHDA is $2Bc_{\max},$ where $c_{\max} \approx 1.$ We can  also show that $p$ must be at least $r.$

## Modifying the analysis of Theorem 1

Apart from the change in the noise injection mechanism, we also note that the bound on the regularization coefficient can be better. Instead of the Gaussian tail bound by Ledoux and Talagrand, we observe that $V = \frac{||\eta \beta\_{-1}||\_2\^2}{\sigma\^2||\beta\_{-1}||\_2\^2}$ is a $\chi^2$ distribution. Using the tail bound of [2] as well as the new noise injection mechanism, the bound on the regularization coefficient in Theorem 1 becomes
$$||\eta\beta\_{-1}||\_2\^2 \leq \frac{8B\^2c\_{\max}\^2r}{\varepsilon\^2} \ln(1.25/\delta)||\beta\_{-1}||\_2\^2.$$

## Modifications in Theorem 2
Due to the change in the noise injection mechanism, the bound on the regularization coefficient  becomes
$$||\eta \beta\_{-1}||\_1 \leq \frac{2B c\_{\max} r \sqrt{2  h_m \ln(1.25/\delta)}}{\varepsilon} ||\beta_{-1}||\_1.$$

The rest of the terms remain same as the previous bound, only $r \log r$ is replaced by $r.$

References:

[1]. The Fast Johnson-Lindenstrauss Transform and Approximate Nearest Neighbors, Ailon and Chazelle, 2009

[2]. Adaptive Estimation of a Quadratic Functional by Model Selection, Laurent and Massart, 2000

---

> ### Author Response · Authors · 2026-04-15
> **Empirical Results**
>
> ## Private Count Sketch
>
> Here, we compare the mean squared errors for the private JL sketch proposed in Sheffet(2017), and the private CountSketch proposed in our paper. Specifically, if the dataset is $(X,y),$ and the sketching matrix is $S,$ then we compute $\beta\_{JL} = \underset{\tilde{\beta}}{\arg\min} ||SA\tilde{\beta}\_{-1}||\_2\^2$(for the private JL sketch,$SA$ being the output of Sheffet's algorithm) or $\beta\_{CS} = \underset{\tilde{\beta}}{\arg \min} ||SHD\hat{A}\tilde{\beta}\_{-1}||\_2\^2,$(for the private Countsketch). Then, we report the mean squared errors $\frac{1}{n}||X\beta - y||\_2\^2$ (the optimal MSE), $\frac{1}{n}||X\beta\_{JL} - y||\_2\^2,$ and $\frac{1}{n}||X\beta\_{CS} - y||\_2\^2$
>
> For our experiments, we take $n = 2\^{15}$ and the number of dimensions to be $10.$ The data matrix $X$ is generated from a standard multivariate normal distribution. We compute $y$ as $y = X\beta\^* + \sigma,$ where $\beta\^* \sim Unif(-1,1)\^{p}$ and $\sigma \sim \mathcal{N}(0,0.5).$ We kept the bound on the $\ell\_2$ norms of the rows as $B=5.$ The sketching dimension was kept at $r = 800$ for all of the experiments. We set the accuracy parameter $\mu = 0.1$ and the failure probability $\nu = 0.01.$ We set $\delta = \frac{1}{n\^2}.$ The privacy budget $\varepsilon$ is varied as shown in the results table. The losses reported in the table are the mean and standard deviations over $10$ independent runs of the experiments. For the experiments, we have  $c_{\max} =  1.07, $ computed from the S and H matrices.
>
>
> ### Comparison of Losses Across $\varepsilon$ Values
>
> | $\varepsilon$ | OLS  | Sheffet's JL Sketch   | Private Count Sketch   |
> |--------:|-----------------------------|-----------------------------------------------|-----------------------------------------|
> | 0.10 | 0.2432 ± 0.0000 | 18.1300 ± 0.1313 | 3.5731 ± 0.2032 |
> | 0.25 | 0.2432 ± 0.0000 | 9.0087 ± 0.0465 | 3.6376 ± 0.1285 |
> | 0.50 | 0.2432 ± 0.0000 | 5.7508 ± 0.0290 | 3.4711 ± 0.1356 |
> | 0.75 | 0.2432 ± 0.0000 | 4.5021 ± 0.0249 | 3.4301 ± 0.1291 |
> | 1.0 | 0.2432 ± 0.0000 | 3.7978 ± 0.0275 | 3.4696 ± 0.1322 |
> | 2.5 | 0.2432 ± 0.0000 | 2.1607 ± 0.0079 | 3.2202 ± 0.1677 |
> | 5.0 | 0.2432 ± 0.0000 | 1.3696 ± 0.0069 | 2.5630 ± 0.1023 |
> | $\infty$ | 0.2432 ± 0.0000 | 0.2471 ± 0.0020 | 0.2476 ± 0.0020 |
>
>
> We observe that for stricter privacy budgets ($\varepsilon \leq 1$), the loss for our private Countsketch method is much lower than that obtained using Sheffet's sketch. This is due to the fact that Sheffet's algorithm requires privately testing the minimum singular value of the dataset, which leads to splitting of the privacy budget. At small $\varepsilon,$ the variance of the noise increases. Due to this, the private singular value test will fail most of the time. Sheffet's algorithm then outputs the sketch of the dataset augmented with $wI.$
> For our method, there is no data dependent test which allows us to utilize the complete privacy budget for privatizing the output sketch. As the privacy budget increases, the losses obtained for  the Sheffet's sketches are lower than our method.
>
>
>
>
> ## Private $\ell_1$ Sketch
>
> We compare the mean absolute errors for sketched $\ell_1$ regression with $\ell_1$ sketch as well as its privatised variant. We solve the sketched problem to compute $\beta = \underset{\tilde{\beta}}{\arg \min}||SA\tilde{\beta}\_{-1}||\_1\^1$ for the non-private sketch and $\beta\_{priv} = \underset{\tilde{\beta}}{\arg \min}||SHD\hat{A}\tilde{\beta}\_{-1}||\_1\^1.$ We report the losses $\frac{1}{n}||X\beta - y||\_1\^1$ and $\frac{1}{n}||X\beta_{priv} - y||\_1\^1,$ aggregated over $10$ independent runs of the experiment.
>
> ### Comparison of Losses Across $\varepsilon$ Values
>
> | $\varepsilon$ | $\ell_1$ Sketch  | Private $\ell_1$ Sketch   |
> |--------:|-----------------------------|-----------------------------------------------|
> | 0.10 | $0.3976 \pm 0.003$ | $4.193 \pm 0.021$ |
> | 0.25 | $0.3976 \pm 0.003$ | $3.102 \pm 0.025$ |
> | 0.50 | $0.3976 \pm 0.003$ | $3.042 \pm 0.016$ |
> | 0.75 | $0.3976 \pm 0.003$ | $1.956 \pm 0.01$ |
> | 1.0 | $0.3976 \pm 0.003$ | $1.78 \pm 0.007$ |
> | 2.5 | $0.3976 \pm 0.003$ | $1.33 \pm 0.009$ |
> | 5.0 | $0.3976 \pm 0.003$ | $0.866 \pm 0.004$ |
> | $\infty$ | $0.3976 \pm 0.003$ | $0.564 \pm 0.007$ |

---

### Decision · Action_Editor_YT5F · 2026-06-29

**Recommendation:** Reject

**Audience:**

Yes

**Audience Explanation:**

Privacy preserving machine learning and statistics is a well represented topic within the TMLR community.

**Claims And Evidence:**

No

**Claims Explanation:**

Reviewers identified two types of concerns with accurate, convincing, and clear evidence.

* One reviewer identified gaps in the proof.  To the best of our knowledge, these gaps have been addressed by the authors in a revision.  Ideally papers should not be submitted with significant gaps that need to be fixed during the review cycle, and the authors spend time before resubmission making sure that proofs are essentially correct.  However, since the fix of the affected lemmas appears to be sound, this was **not** part of the justification for the reject decision.

* Multiple reviewers pointed out issues with placing of the paper that impact the thoroughness of the evaluation.  Whether the paper is claiming a theoretical or a practical improvement, the paper should be revised to give a thorough comparison to other methods for private regression, rather than a small set of approaches most similar to the one the authors chose.

**Resubmission Of Major Revision:**

The authors may consider submitting a major revision at a later time.